# VCODE: A MULTIMODAL CODING BENCHMARK WITH SVG AS SYMBOLIC VISUAL REPRESENTATION

## ABSTRACT

Code has emerged as a precise, executable medium for reasoning and action in the agent era. Yet progress has largely focused on linguistic-centric tasks (*e.g.,* program synthesis, debugging), leaving *visual-centric coding* underexplored. Conventional image representations rely on dense RGB pixels that capture appearance but provide limited symbolic abstraction. Inspired by how humans reason over sketches, we advocate SVG code as a compact, interpretable, and executable visual representation. We introduce **VCode**, a benchmark that reframes multimodal understanding as code generation: given an image, a model must produce SVG that preserves symbolic meaning for downstream reasoning. VCode covers three challenging domains—general commonsense (MM-Vet), professional disciplines (MMMU), and visual-centric perception (CV-Bench). To assess symbolic fidelity, we propose **CodeVQA**, a novel evaluation protocol in which *a policy model answers questions over rendered SVG; correct answers indicate faithful symbolic preservation*. Empirically, frontier VLMs struggle to generate faithful SVGs, revealing a persistent gap between language-centric and visual-centric coding. To close this gap, we introduce **VCoder**, an agentic framework that augments VLMs along two axes: (i) *Thinking with Revision*, which iteratively analyzes discrepancies and refines SVG code; and (ii) *Acting with Visual Tools*, where detectors and parsers supply structured cues (objects, shapes, text) beyond intrinsic model capacity. Across benchmarks, frontier VLMs with strong reasoning score well overall yet remain limited on professional knowledge and 3D reasoning; VCoder delivers a +8.7-point overall gain over the top-performing Claude-4-Opus. Human studies further show that although VLMs score higher on raw images, humans are more robust on rendered SVGs—underscoring symbolic visual coding as a promising paradigm for human-like multimodal intelligence.

## 1 INTRODUCTION

To advance reasoning and agentic intelligence, code has emerged as a powerful medium for interacting with digital environments Park et al. (2023); Wang et al. (2023); Liu et al. (2023). Unlike natural language, which is free-form and descriptive, code is precise, structured, and executable—making it an effective mechanism for action. Consequently, recent benchmarks have predominantly emphasized *linguistic-centric* coding abilities, covering tasks such as program synthesis, debugging, and competitive programming Chen et al. (2021); Austin et al. (2021); Jain et al. (2024); Tian et al. (2024); Jimenez et al. (2023), where success is measured by both correctness and executability. In the multi-modal regime, coding plays a crucial role in generating executable programs that interface with tools or environments to accomplish complex task, a paradigm that has gained particular traction in embodied agents Wang et al. (2023); Liang et al. (2022). A parallel line of work leverages code to generate *synthetic* visual assets—such as charts Yang et al. (2024a); Wu et al. (2024), diagrams Rodriguez et al. (2025); Chen et al. (2025), or websites Beltramelli (2018); Si et al. (2024)—which synthesis assets, are not directly grounded in the natural visual world.

When recapping the representation for natural images, the dominant practice has been to encode them as pixels or superpixels. These representations are effective in that they densely capture visual apperance. In contrast, humans often perceive and reason through sparse symbolic sketches that emphasize spatial relationships, object counts, and structural outlines Hu et al. (2024). Similar to an

Figure 1: **Illustration of VCode.** An RGB image (left, represented by pixels) is translated into symbolic SVG code (middle) via VLM as Coder and rendered back into an image (right, represented by code), aiming to preserve symbolic meaning (*e.g.,* "three sheep on the farm"). As shown at the bottom, VCode provides a compact, interpretable, and executable representation of original images.

artist drafting a rough sketch before filling in appearance details, such abstraction offers a compact yet informative scaffold for reasoning. Building on this intuition, we propose using Scalable Vector Graphics (SVG) code as an alternative visual representation, owing to its compact, interpretable, and executable nature. Thus, SVGs have long been used for icons and logos Wu et al. (2023); Rodriguez et al. (2025); Yang et al. (2025) for a general visual abstraction. This perspective motivates a fundamental question: *can visual representation move beyond raw pixels and learn to represent and reason through code?*

In this work, we introduce VCode, a multimodal coding benchmark that pioneers the use of SVG code as a visual representation. VCode is constructed by repurposing existing multimodal understanding benchmarks across three domains: General commonsense (MM-Vet Yu et al. (2024)), College-level disciplines (MMMU Yue et al. (2024)), and Visual-centric Perception (*e.g.,* 3D depth and relationships in CV-Bench Tong et al. (2024)). VCode reframes these tasks as visual coding: given an image, a model must generate SVG code that faithfully renders the image, thereby reconstructing its symbolic representation. To evaluate this transformation, we propose **CodeVQA**, a novel protocol in which a vision–language model must answer core questions about the original image by reasoning over the rendered SVG. This provides a principled test of *whether the generated code serves as an adequate and faithful visual representation*. Experiments on VCode show that existing coders remain limited in such challenging setting. We observe that coding quality improves with a model's reasoning ability, yet models still fail to preserve fine-grained visual relations (*e.g.,* near vs. far), exposing a persistent gap between language- and visual-centric coding. Notably, human studies further show that although VLMs score higher on raw images, humans are more robust when reasoning over rendered SVGs.

To this end, we augment existing coders with two complementary capabilities. **(i) Thinking with Revision.** The model compares intermediate renderings with the original image, explicitly articulates discrepancies, and iteratively updates the SVG to improve fidelity. **(ii) Acting with Visual Tools.** We equip the coder with external perception toolboxes—*e.g.,* object detectors and segmenters Xiao et al. (2023); Ravi et al. (2024)—to supply structured cues (objects, shapes, text) as coding context. Together, these strategies yield a +8.7 overall gain over the top-performing Claude-4-Opus, substantially strengthening visual-centric coding. Our contributions are threefold:

1. **VCode: A Novel Perspective for Multimodal Coding.** We recast multimodal understanding as *visual-centric coding*: given an image, generate SVG that preserves symbolic structure for downstream reasoning. We further present **CodeVQA** – a protocol that asks a VLM to answer the *original-image* questions using only the *rendered SVG*, thereby testing whether the code is an adequate and faithful visual representation.

2. **VCoder: Augmenting VLM as Strong Multimodal Coders** via *(i) Thinking with Revision* (iterative discrepancy analysis and SVG refinement) and *(ii) Acting with Visual Tools* (structured visual cues from detectors). VCoder achieves a significant overall gain over a strong baseline.

3. **Evaluation and Insights.** Extensive experiments expose persistent weaknesses of frontier VLMs in visual-centric coding. Human studies show greater robustness when reasoning over rendered SVGs than raw images, suggesting symbolic visual coding as a promising path advancing human-like multimodal intelligence.

| Benchmarks | Domain | Size | Inputs | Outputs | Evaluation |
|---|---|---|---|---|---|
| *Coding* | | | | | |
| HumanEval Chen et al. (2021) | Algorithm | 164 | Text | Code | Execute Pass |
| MMCode Li et al. (2024) | Visualization | 263 | Text & Img | Code | Execute Pass |
| ChartMimic Yang et al. (2024a) | Chart | 4800 | Text & Img | Code | Similarity |
| Design2Code Si et al. (2024) | Web UI | 484 | Text & Img | Code | Similarity |
| SWE-Bench Jimenez et al. (2023) | GitHub | 2294 | Text & Code | Code | Execute Pass |
| SVG-Bench Rodriguez et al. (2025) | SVG | 23K | Img / Text | Code | Similarity |
| *Multi-modal* | | | | | |
| MM-Vet Yu et al. (2024) | General | 218 | Img. & text | Text | OpenQA |
| MMBench Liu et al. (2024) | General | 3217 | Img. & text | Text | MCQ |
| MMMU Yue et al. (2024) | College | 11.5K | Img. & text | Text | OpenQA / MCQ |
| MMMU-Pro Yue et al. (2025) | College | 1730 | Img. & text | Text | OpenQA / MCQ |
| CV-Bench Tong et al. (2024) | Perception | 2638 | Img. & text | Text | MCQ |
| **VCode (Ours)** | **G&C&P** | 464 | Img. | Code | **Render→VQA** |

Table 1: **Comparison of VCode with coding (top) and multimodal (bottom) benchmarks.** VCode differs in three ways: **(i) Task**: models must generate *code* directly from natural images, without extra query guidance; **(ii) Scope**: focuses on natural multimodal understanding across diverse domains—General (G), College (C), and Perception (P); **(iii) Evaluation**: introduces **CodeVQA** (Render → VQA), which judges whether the rendered SVG preserves the original image's symbolic meaning.

## 2 RELATED WORKS

### 2.1 CODING BENCHMARKS

**Coding in LLMs.** Despite there being several coding benchmarks, most of them are initially developed for purely language coding. Representative efforts include HumanEval Chen et al. (2021) and MBPP Austin et al. (2021), which evaluate correctness of synthesized programs given natural language or code-level prompts. Later benchmarks such as SWE-Bench Jimenez et al. (2023) extend this paradigm to real-world software engineering, requiring models to resolve issues directly in large GitHub repositories. Despite their diversity, these benchmarks are fundamentally **linguistic-centric**: the inputs and outputs remain in textual or code form, with success measured by pass rates or test-case execution. While effective in quantifying reasoning over program text, such settings offer little insight into multimodal capabilities.

**Coding in Multi-modal.** Moving beyond purely textual code, a line of work incorporates visual observations into coding tasks. Benchmarks such as Plot2Code Wu et al. (2024),Design2Code Si et al. (2024), and ChartMimic Yang et al. (2024a) translate charts or UI mockups into executable plotting or layout code. MMCode Li et al. (2024) and SWE-Bench-MM Yang et al. (2024b) further integrate images alongside text, exploring how multimodal inputs can inform code generation. At larger scale, SVGenius Chen et al. (2025) (generation, editing, understanding) evaluates models' ability to produce vector-graphic code, highlighting challenges in preserving both semantics and structure. Despite this progress, most of these datasets emphasize **synthetic** visual assets (*e.g.,* charts, Web UI, vector icons) as shown in Tab.1 top-half, leaving open the question of whether models can encode real-world natural images into executable visual code. This gap motivates our VCode benchmark, which repurposes multimodal understanding tasks into the visual coding with SVG.

### 2.2 MULTIMODAL UNDERSTANDING

Various benchmarks systematically evaluate multimodal understanding. Early efforts such as MM-Bench Liu et al. (2024) and MM-Vet Yu et al. (2024) emphasize general perception and text–image reasoning. More recent benchmarks, including MMMU Yue et al. (2024) and MMMU-Pro Yue et al. (2025), target professional knowledge and domain-specific reasoning. However, most of these evaluations interact with models through *natural language* (*e.g.,* query or answer). In VCode, we argue that generating *code* to represent natural images constitutes an even more advanced form of understanding. As illustrated in Tab.1 bottom-half, unlike traditional perception tasks, this requires the model to distill an image into its core concepts and structural features by a *render* image, and to express them in a symbolic format that bridges perception with reasoning and action.

## 3 VCODE BENCHMARK

### 3.1 TASK DEFINITIONS

As illustrated in Fig.1, given an input RGB image $\mathcal{V}$, a vision–language model $\psi$ is tasked with generating SVG code $\mathcal{C}$ that encodes the image. Rendering this code yields a rendered image $\widetilde{\mathcal{V}}$. The objective is to minimize the discrepancy between the symbolic information of the original and rendered images:

$$\mathcal{L} = \min \left| I(\mathcal{V}) - I(\widetilde{\mathcal{V}}) \right|, \tag{1}$$

where $I(\cdot)$ denotes a symbolic information representation. The central challenge, however, lies in defining an applicable measure of symbolic information, which we elaborate on below.

### 3.2 EVALUATION METRICS

The key to the evaluation prototype lies in how the correspondence between the input image $\mathcal{V}$ and the rendered image $\widetilde{\mathcal{V}}$ is defined.

**SigLip Score.** To define what constitutes an ideal SVG representation, we argue that it should faithfully preserve the semantic content of the original image rather than merely matching pixel-level similarity. One way to measure this is through embedding consistency. We leverage a pretrained visual encoder $f(\cdot)$ such as SigLIP Zhai et al. (2023); Tschannen et al. (2025) to extract embeddings for both $\mathcal{V}$ and $\widetilde{\mathcal{V}}$, and compute their cosine distance:

$$\mathcal{L} = \max \cos \left( f(\mathcal{V}), f(\widetilde{\mathcal{V}}) \right). \tag{2}$$

**CodeVQA.** A more direct criterion is whether the rendered image $\widetilde{\mathcal{V}}$ alone supports correct reasoning. Usually, $\widetilde{\mathcal{V}}$ may even facilitate answering questions that are ambiguous or harder to resolve from the original $\mathcal{V}$. Hence, the evaluation should not be constrained by the original image's responses, but instead focus directly on the correctness of answers derived from $\widetilde{\mathcal{V}}$. We define a *policy* model $\phi$ that outputs an answer $\mathcal{A}$ given an image and a question $\mathcal{Q}$. Then goal is formulated as

$$\mathcal{A} = \phi \left( \widetilde{\mathcal{V}}, \mathcal{Q} \right),$$
$$\mathcal{L} = \max \mathbf{1}[\texttt{Evaluator}(\mathcal{A})]. \tag{3}$$

where $\mathbf{1}[\cdot]$ is the indicator function. $\texttt{Evaluator}(\cdot)$ is a rule-based matching in multiple-choices setting, and it can be a LLM-as-Judge in open-ending. If the answer is correct, the SVG suffices to convey the required semantics; otherwise, it reveals a gap in representational fidelity.

**Code tokens.** Beyond faithful representation, we argue that an effective coder should represent an image with as few code tokens as possible, producing a concise yet faithful representation. To assess this efficiency, we evaluate the length of the generated SVG in terms of its token count $|\mathcal{C}|$.

### 3.3 DATA CURATION

With the evaluation prototype in place, the next step is to develop appropriate question sets $\mathcal{Q}$ for each associated image $\mathcal{V}$. To this end, we repurpose existing multimodal visual question answering benchmarks to align with our objective. To ensure diversity in taxonomy and difficulty, we focus on three representative scenarios: **(i) Commonsense perception**: Assesses a model's ability to capture everyday semantics such as spatial relationships. We adopt MM-Vet Yu et al. (2024) as the source. **(ii) Professional knowledge**: Targets domain-specific, diploma-level tasks that demand both reasoning and coding skills. We incorporate the development set of MMMU Yue et al. (2024), which spans multiple disciplines and requires deeper reasoning and expert knowledge. **(iii) Visual-centric**: Evaluates performance in visually intensive settings involving counting, distance estimation, and relative spatial relationships in 2D or 3D. We draw from CV-Bench Tong et al. (2024).

**Data statistics.** Following this three-pronged curation strategy, we processed each source benchmark to construct our final dataset. For (i) commonsense perception, we incorporated the entirety of MM-Vet Yu et al. (2024), resulting in 218 image-question pairs. For (ii) professional knowledge, our

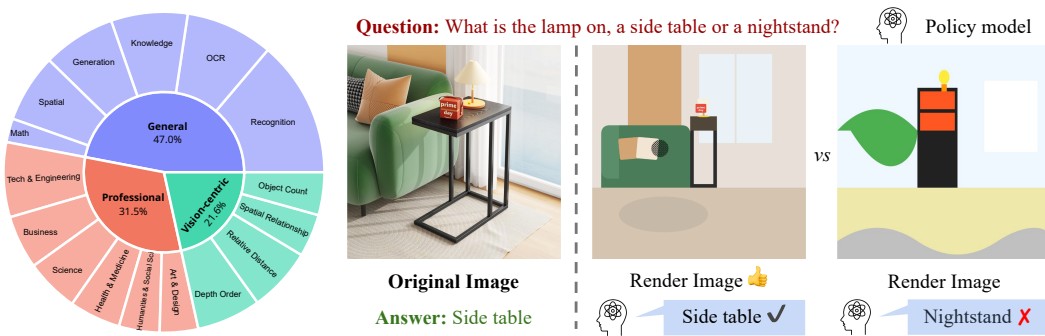

(a) **Distributions of VCode.**    (b) **Illustration of CodeVQA prototype.**

Figure 2: **Left: Distributions of tasks in VCode**, showing the proportions of general, professional, and vision-centric categories. **Right: Illustration of the CodeVQA prototype**: given an image and a question (*e.g.,* "What is the lamp on, a side table or a nightstand?"), the policy model answers based on the rendered image. A correct answer indicates that the SVG representation preserves the semantic content of the original image, while an incorrect answer highlights room for improvement.

curation involved filtering the MMMU Yue et al. (2024) development set to retain only single-image VQA instances, which yielded a specialized subset of 146 pairs. Finally, for (iii) visual-centric, we created a balanced 100-pair subset from CV-Bench Tong et al. (2024) through a stratified sampling process. This involved shuffling the data and applying interval selection to ensure a specific distribution across its sub-tasks: spatial relationship (20), object count (20), depth order (30), and relative distance (30). In total, this process yields 464 image-question pairs. The taxonomy distribution of VCode is illustrated in Fig. 2(a).

## 4 VCODER

In practice, we find that directly prompting Coders to generate SVG code from natural images remains highly challenging. This difficulty arises from three factors: **(i) Long-Context Code Inputs:** fully representing an image typically requires thousands of tokens; composing such long sequences demands strong code reasoning over complex elements, beyond what current Coders provide. **(ii) Visually-Blind Outputs:** inputs and outputs are cross-modal; because the rendered image is unseen until execution, the model must anticipate the visual consequences of code edits during generation. **(iii) Weak Visual Fineness:** for irregular objects (*e.g.,* a dog's boundary), language models struggle to capture low-level details—edges, masks, and colors—that must be encoded precisely as numeric values, even though these are fundamental to code-based representations.

To address these intertwined challenges, we propose augmenting Coders at test time with two complementary capabilities. **(a) Thinking with Revision:** we enhance reasoning through test-time scaling and a revision strategy that allows the model to iteratively refine its outputs, bridging the gap between long-context code generation and faithful visual rendering. **(b) Acting with Vision Tools:** we equip Coders with external tools that extract fine-grained visual cues—such as edges, masks, and color regions—and translate them into structured code signals, enabling models to overcome their inherent limitations in low-level perception.

### 4.1 THINKING WITH REVISION

Since the initial reconstruction may not always yield a satisfactory result, a natural way to enhance Coders is to let them re-examine their own outputs and iteratively refine the code. Our revision strategy follows a two-step loop: detect discrepancies between the rendered output and the target image, then update the code conditioned on these differences.

(*i*) **Comment the Difference.** Given an intermediate rendering $\widetilde{\mathcal{V}}^{(t)}$, the coder first perceives its deviation from the original image $\mathcal{V}$. Although VLMs may be limited as Coders, they are already strong in visual perception. We therefore design the revision process to let them capture differences

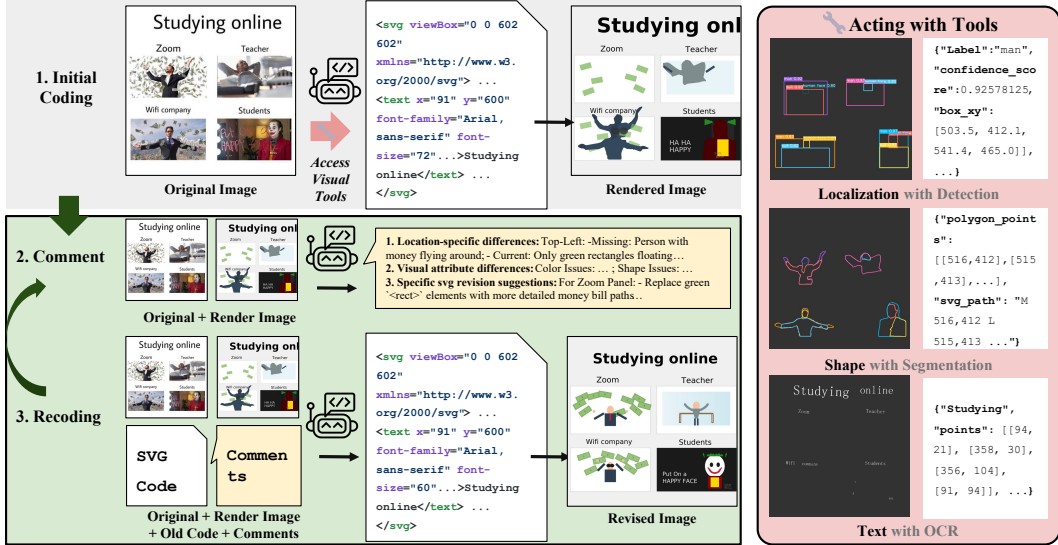

Figure 3: **Augmenting Coders with Test-time Revision & Visual Tools. Left:** Thinking with Revision – the model performs initial coding, comments on discrepancies between original and rendered images, and iteratively refines the SVG code. **Right:** Acting with Vision Tools – external modules provide cues on categories, locations, shapes, colors, and text, which are translated into structured code signals to guide generation. These techniques yield more faithful and accurate renderings.

through two observations. At each iteration $t$, we compute a difference signal $\Delta^{(t)} \leftarrow \psi\big(\mathcal{V}, \widetilde{\mathcal{V}}^{(t)}\big)$, which quantifies the discrepancy between the reconstruction and the target.

**(ii) Revise with Render Feedback.** The difference signal $\Delta^{(t)}$, together with the current code $\mathcal{C}^{(t)}$ and render $\widetilde{\mathcal{V}}^{(t)}$, is provided to the coder $\psi$ to generate revised code $\mathcal{C}^{(t+1)}$. Executing this code produces an updated reconstruction $\widetilde{\mathcal{V}}^{(t+1)} \leftarrow \big(\mathcal{V}, \widetilde{\mathcal{V}}^{(t)}, \mathcal{C}^{(t)}, \Delta^{(t)}\big)$.

This revision loop is repeated for $t = 0, 1, \ldots, T$, progressively refining the reconstruction until a satisfactory visual outcome is reached. The full procedure is summarized in Algorithm 1.

## 4.2 ACT WITH VISUAL TOOLS

Another limitation for Coder is capture the image fine-grained attribution such as boundary. Here we able the Coder to access additional visual tools to provide meta information to complement the generated SVG quality. We display part of tools with supple information in the right side of Fig.3.

**Category.** Object categories are obtained from a detector Xiao et al. (2023) and provide the Coder with essential semantic labels. For example, a detected object can be annotated in SVG with an attribute like `id='bird'`. These labels serve as the basic prior for generation and are always combined with geometric cues like location or shape to describe each object more completely.

**Location.** A key factor in reconstruction is capturing where objects appear in the image. To provide this information, we rely on bounding boxes predicted by Florence-2 Xiao et al. (2023), expressed as absolute coordinates $(x_1, y_1, x_2, y_2)$ together with the image width and height. These cues allow the Coder to anchor elements at the correct positions on the canvas, preserving the overall layout.

**Shape.** While regular geometric primitives are straightforward to express, a key challenge lies in representing irregular boundaries. To address this, we employ SAM-2 Ravi et al. (2024) to generate segmentation masks that capture detailed object contours. These masks are then downsampled into sparse coordinate points through an adaptive simplification strategy, which reduces the number of vertices while keeping the overall area nearly unchanged. The resulting polygonal paths provide the Coder with compact yet faithful shape descriptions that complement category and location cues.

**Text.** Text often carries critical semantic information that cannot be replaced by shapes or colors. To incorporate this, We apply OpenOCR Du et al. (2025) to detect and transcribe text regions, and directly encode them into SVG using the native `<text>` tag, which preserves both content and visual attributes without the rendering issues of pixel-based methods.

| Model name | SigLip score | Code token ($K$) | CodeVQA | | | | | | | MMMU Avg. | CV-Bench | | | Overall |
| | | | Rec | Ocr | Know | MM-Vet Gen | MM-Vet Spat | Math | Avg. | | 2D | 3D | Avg. | |
| Orig. Image (4o-mini) | 100. | N/A | 60.5 | 78.9 | 58.5 | 59.5 | 70.9 | 84.2 | 67.1 | 47.9 | 43.1 | 61.7 | 52.4 | 55.8 |
| Claude-4-Opus | 65.6 | 1.5 | 30.4 | 52.3 | 13.9 | 18.5 | 49.5 | 50.4 | 37.5 | 43.2 | 38.0 | **60.0** | 49.0 | 41.8 |
| Claude-4-Sonnet | 65.2 | 1.6 | 31.8 | 51.2 | 24.9 | 27.9 | 44.8 | 34.6 | 37.8 | 41.8 | 42.5 | 48.3 | 45.4 | 40.7 |
| GPT-5 | **72.1** | 2.3 | **33.9** | **64.9** | 20.5 | 23.8 | 60.5 | 65.4 | 43.9 | 42.5 | **53.3** | 55.0 | **54.2** | 45.7 |
| GPT-4o | 60.7 | 0.6 | 23.1 | 58.4 | 12.7 | 17.0 | 51.3 | 60.4 | 35.0 | 44.5 | 23.4 | 43.3 | 33.4 | 37.6 |
| GPT-o3 | 63.6 | 1.1 | 31.3 | 55.2 | 17.7 | 19.7 | 48.5 | 61.5 | 39.8 | 39.0 | 38.7 | 51.7 | 45.2 | 40.7 |
| GPT-4.1 | 68.5 | 1.6 | 30.8 | 62.0 | 15.5 | 20.4 | 56.0 | 55.8 | 40.9 | 44.5 | 43.9 | 61.7 | 52.8 | 44.6 |
| GPT-4o-mini | 62.5 | 0.4 | 20.7 | 58.4 | 13.2 | 18.9 | 46.8 | 63.5 | 33.5 | 44.5 | 23.4 | 48.3 | 35.9 | 37.5 |
| Gemini-2.5-Pro | 66.4 | 2.4 | 28.9 | 57.8 | 20.0 | 22.9 | 47.9 | 68.5 | 39.1 | 45.2 | 48.2 | 50.0 | 49.1 | 43.2 |
| Gemini-2.5-Flash | 63.6 | 1.9 | 29.3 | 56.7 | 17.4 | 21.1 | 46.3 | 53.8 | 39.1 | 39.7 | 32.1 | 53.3 | 42.7 | 40.1 |
| Seed-1.6-thinking | 62.7 | 1.6 | 18.9 | 46.5 | 8.1 | 11.9 | 44.1 | 38.5 | 28.7 | 43.2 | 36.6 | 51.7 | 44.1 | 36.6 |
| Qwen2.5-VL-72B | 58.9 | 0.6 | 20.6 | 52.9 | 14.0 | 17.3 | 51.3 | 43.1 | 31.8 | 41.8 | 20.5 | 53.3 | 36.9 | 36.0 |
| Qwen2.5-VL-7B | 24.4 | 0.3 | 4.9 | 6.0 | 3.0 | 4.0 | 7.1 | 3.8 | 4.8 | 34.9 | 13.2 | 36.7 | 24.9 | 18.6 |
| InternVL3.5-241B-A28B | 59.6 | 1.0 | 20.4 | 52.4 | 11.9 | 15.7 | 39.2 | 42.3 | 31.1 | 43.8 | 22.6 | 48.3 | 35.5 | 36.0 |
| Intern-S1 | 59.3 | 1.0 | 24.7 | 56.8 | 12.1 | 16.0 | 51.2 | 41.9 | 35.2 | 41.8 | 36.6 | 48.3 | 42.5 | 38.9 |
| InternVL3-78B | 58.0 | 0.7 | 16.9 | 52.7 | 8.3 | 13.9 | 40.5 | 55.0 | 29.1 | 45.2 | 19.1 | 45.0 | 32.0 | 34.8 |
| GLM-4.5V | 63.4 | 1.6 | 22.4 | 54.4 | 7.1 | 15.6 | 46.0 | 56.9 | 33.1 | 40.4 | 16.9 | 43.3 | 30.1 | 34.8 |
| GLM-4.1V-Thinking | 62.1 | 1.2 | 21.1 | 52.0 | 10.4 | 13.7 | 44.8 | 58.8 | 31.9 | 43.2 | 32.2 | 48.3 | 40.3 | 37.3 |
| MiniCPM-V-4.5 | 46.2 | 0.9 | 11.8 | 31.8 | 4.5 | 10.8 | 23.2 | 26.5 | 17.7 | 39.5 | 27.7 | 53.3 | 40.5 | 29.5 |
| StarVector-8B | 5.1 | 1.3 | 0.0 | 3.4 | 0 | 1.6 | 4.4 | 0 | 1.5 | 26.0 | 0 | 0 | 0 | 8.9 |
| **VCoder (Claude-4-Opus)** | 71.1 | 2.0 | **46.0** | 64.4 | **40.8** | **43.0** | **61.6** | 72.7 | **54.1**$_{+16.6}$ | **50.0**$_{+6.8}$ | 43.9 | 43.3 | 43.6 | **50.5**$_{+8.7}$ |

Table 2: **Main results on VCode** across various top-performing VLM coders. The best scores are in **bold**.

## 5 EXPERIMENTS

### 5.1 BASELINE AND SETTINGS

To comprehensively evaluate our proposed framework, we compare it against a wide range of proprietary and open-source models that represent the current state of the art in multimodal reasoning and code generation. *Proprietary models,* such as Claude-4-Opus and Claude-4-Sonnet, GPT-5, GPT-4.1, GPT-o3, GPT-4o, and GPT-4o-mini Hurst et al. (2024), as well as Gemini-2.5-Pro and Gemini-2.5-Flash Comanici et al. (2025), and Seed-1.6-thinking. These models are widely recognized for their strong reasoning and multimodal capabilities, and thus provide competitive upper baselines for our benchmark. *Open-source models:* including Qwen2.5-VL-72B and Qwen2.5-VL-7B Team (2025), InternVL3.5-241B-A28B Wang et al. (2025), Intern-S1, InternVL3-78B Zhu et al. (2025), MiniCPM-V-4.5 Yao et al. (2025), GLM-4.5V and GLM-4.1V-Thinking Team et al. (2025), and StarVector Rodriguez et al. (2025). These baselines cover a diverse spectrum of model sizes and training paradigms, enabling a comparison between proprietary and open-source approaches.

**Evaluation settings.** Unless otherwise noted, all models are queried under a unified prompting interface with identical inputs to ensure fairness. The primary automatic evaluator is GPT-4o-mini, which provides consistent judgments across benchmarks.

### 5.2 MAIN RESULTS

In Tab. 2, we evaluate full baselines on VCode, reporting per-domain results—general, college, and vision-centric—and the overall average. We have the following observation.

**Stronger reasoning yield better visual coding scores.** Closed-source models consistently outperform open-source counterparts across benchmarks. GPT-5 sets the strongest baseline with the top SigLip score (72.1) and the highest CodeVQA overall (46.2), showing robust performance on both similarity and reasoning metrics. This pattern indicates that stronger reasoning ability translates into better VCode performance—*i.e.,* models that reason well produce more faithful symbolic renderings. We also observe a positive correlation between semantic similarity (SigLip) and CodeVQA.

**Challenges across different dimensions.** *(i) Best performer still trails the original-image upper bound.* Even the best SVG result—GPT-5 at 45.7—remains well below the raw-image upper bound (55.8), indicating substantial headroom. This confirms that the task is sufficiently challenging and that symbolic representation still has ample room for improvement. *(ii) SVG specialist underperforms.* StarVector-8B ranks last, highlighting VCode's difficulty and the gap between neatly authored SVG corpora and SVGs derived from natural images. *(iii) Knowledge is hardest.* The Know dimension is consistently the lowest, reflecting the compounded challenge of recalling facts and then encoding them faithfully in SVG (*e.g.,* historical entities). *(iv) Professional disciplines are hard to*

*differentiate.* On MMMU, models cluster within a narrow, modest band, and most fail the more demanding disciplinary settings. *(v) Vision-centric perception is tough.* CV-Bench scores hover near the low (randomly by 50%), especially on 3D relations (depth or spatial). Even with VCoder, improvements are meaningful but leave substantial headroom.

**Absolute gains with VCoder.** Built on Claude-4-Opus, VCoder lifts Overall from 41.8 to 50.5 (+8.7) via revision and vision-tool assistance, improving all three domains—demonstrating an effective enhancement for visual-centric coding.

**Code token length.** Models that emit short SVGs underperform (*e.g.,* Qwen-2.5-VL), suggesting under-specification. By contrast, stronger models (GPT-5, Gemini-2.5-Pro) produce substantially longer sequences (often > 2K tokens) and attain higher scores. Length is not sufficient on its own, but performance scales with usable context, highlighting long-context reasoning and generation as a central bottleneck for visual-centric coding.

## 5.3 KEY ABLATIONS

**Effects of VCoder modules.** Ablations in Tab. 3 reveal three takeaways: *(i)* Adding fine-grained cues (location, category, shape) yields steady gains; shape is especially helpful for spatial reasoning (Spat.), even without large changes in SigLip, indicating structural benefits. *(ii)* Text cues help, with the full visual-tool ensemble provides the largest overall improvement. *(iii)* Test-time Revision adds a further boost on top of the tools—most notably on knowledge, generation, and math—showing that structured cues and iterative refinement are complementary.

| Variant | SigLip score | \multicolumn{7}{c}{CodeVQA} | | | | | | |
|---------|--------------|-----|-----|------|-----|------|------|------|
| | | Rec | Ocr | Know | Gen | Spat | Math | **Avg.** |
| Claude-4-Opus | 65.6 | 30.4 | 52.3 | 13.9 | 18.5 | 49.5 | 50.4 | 37.5 |
| +Loc. & *C.* | 70.8 | 29.7 | 60.3 | 17.5 | 22.9 | 54.9 | 46.2 | 39.7 |
| +Loc. & *C.* & *S.* | 71.5 | 33.4 | 62.7 | 19.3 | 25.1 | 63.1 | 64.2 | 43.3 |
| +Text | 69.9 | 30.4 | 59.5 | 19.2 | 21.5 | 56.8 | 65.4 | 41.5 |
| +Visual tool | **71.6** | 44.5 | **66.0** | **34.9** | **38.9** | **67.7** | 61.5 | 52.4 |
| VCoder(+Revision) | **71.6** | **46.0** | 64.4 | **40.8** | **43.0** | 61.6 | **72.7** | **54.1** |

Table 3: **Effects by by VCoder modules,** where Loc. denotes Location, *C.* denotes Category, and *S.* denotes Shape.

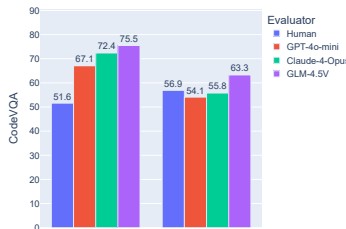

Figure 4: **Effects by different policy during evaluation**

**Effects across policies and Human studies.** Fig. 4 shows the performance differences across policy $\phi$, including humans. On the original images, all models substantially surpass human perception and reasoning (51.6 for humans vs. 75.5 for GLM-4.5V). However, when evaluated on SVG representations, all models exhibit a noticeable performance drop, whereas the human score increases (56.9). This *highlights humans' strong capacity for symbolic understanding*, enabling them to outperform advanced models such as GPT-4o-mini and Claude-4-Opus in this setting. The divergence suggests current VLMs are over-tuned to appearance cues, while humans benefit from the abstraction; improving model robustness to symbolic inputs is therefore a promising direction.

**Effects by Revision.** In Fig. 5, we examine the impact of the test-time revision strategy. Both Claude and GLM-4.5V benefit from revision, with GLM-4.5V showing the most substantial gains—likely due to its built-in "thinking mode," which excels at difference analysis and refinement. By contrast, GPT-4o struggles to leverage revision effectively, reflecting its limitations in deeper reasoning.

| Variant | SigLip | Code Token | \multicolumn{4}{c}{Code2QA} | | | |
|---------|--------|------------|--------|------|----------|-----------|
| | | | MM-Vet | MMMU | CV-Bench | **Overall** |
| Img2SVG | 65.6 | 1.5K | 37.5 | **43.2** | 52.3 | 42.5 |
| Img2Text2Svg | 68.5 | 1.8K | **43.0** | **43.2** | 55.6 | **46.4** |
| Img2Svg* | 69.8 | 1.6K | 38.2 | 42.5 | 53.7 | 43.5 |

Table 4: **Effects by different input modes.** There *denotes we run this method with *thinking-mode*.

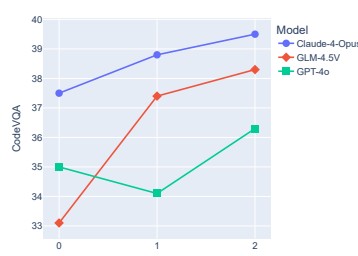

Figure 5: **Effect of Thinking revision.**

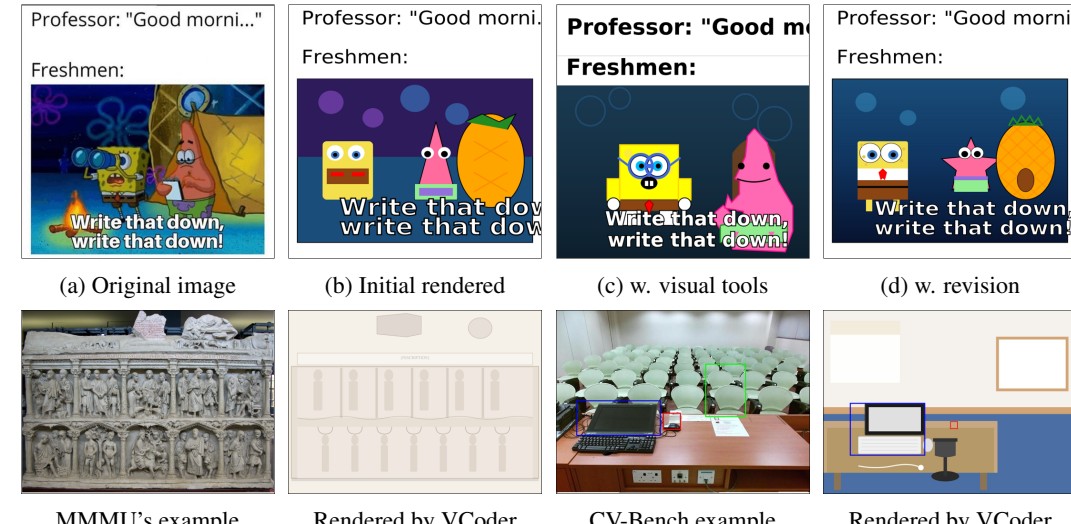

| (a) Original image | (b) Initial rendered | (c) w. visual tools | (d) w. revision |

| MMMU's example | Rendered by VCoder | CV-Bench example | Rendered by VCoder |

Figure 6: **Qualitative examples from VCode. Top row (a–d):** an internet meme rendered progressively by initial decoding, visual-tool assistance, and revision. **Bottom row:** challenge samples from MMMU (Art-Theory) and CV-Bench (3D), alongside their SVG renderings by VCoder.

**Effects by Visual v.s. Textual query.** In Tab. 4, we examine the impact of input modality. Using raw images (*i.e.,* Img2SVG) gives the weakest results, suggesting that current coders are poorly adapted to direct visual input. By contrast, translating images into linguistic captions before coding (*i.e.,* Img2Text2Svg) achieves the best performance, highlighting the benefit of language as an intermediate representation. Notably, even with deep thinking enabled (*i.e.,* Img2Svg-Thinking), performance remains low, underscoring the difficulty of visual-centric coding and the gap between language-driven and vision-driven code generation.

## 5.4 QUALITATIVE ANALYSIS

Fig. 7 presents qualitative results by comparing origina image and the rendered image by VCoder. **Top row.** Across the four stages, the initial decoding misses layout and semantics. Adding *visual tools* recovers key geometry (*e.g.,* the starfish character's triangular body and facial features), while *revision* corrects fine details (character proportions, text alignment, spacing), yielding a rendering that closely matches the meme's structure and intent. **Bottom row.** VCoder produces SVGs that are both more faithful to the source and more interpretable for downstream reasoning. The left example (MMMU) is knowledge-intensive: accurately depicting a multi-panel historical relief requires domain cues and fine structural abstraction, where base models often collapse detail. The right example (CV-Bench) is vision-centric: success hinges on *visually grounded prompts* that localize and size objects correctly (*e.g.,* monitor in front of keyboard, receding rows of chairs), after which revision tightens residual misalignments. These examples underscore the challenges by VCode.

## 6 CONCLUSION

We introduced VCode, offering a new perspective on multimodal coding by benchmarking multimodal understanding with SVG as a visual representation, along with CodeVQA to assess symbolic fidelity through QA over rendered SVGs. Our study shows that frontier VLMs struggle to produce faithful SVGs despite strong linguistic reasoning, revealing a persistent gap between language- and vision-centric coding. To address this, we proposed VCoder, which integrates Test-time Revision and Acting with Visual Tools, yielding substantial improvements. Human studies further indicate that people are more robust on rendered SVGs than on raw images, underscoring the promise of symbolic visual coding for multimodal intelligence. Future work can focus on developing end-to-end vision–language coders with scalable training data to enable more faithful symbolic representations.

ETHICS STATEMENT

This work repurposes existing public multimodal datasets (MM-Vet, MMMU, CV-Bench) and does not involve sensitive or private information. Human evaluation was conducted with informed consent and fair compensation. We see minimal risk of harm; potential misuse (e.g., generating misleading visualizations) is noted, and we release our benchmark strictly for research purposes.

REPRODUCIBILITY STATEMENT

We provide details of dataset construction, evaluation protocols, and model settings in the main text and appendix. All datasets used are publicly available, and our benchmark, code, and evaluation scripts will be released upon publication to facilitate replication of our results.

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

## A   IMPLEMENT DETAILS

We implement our model using the PyTorch framework on an NVIDIA RTX 4090 GPU with 24GB of memory. The maximum output length is set to 16,384 tokens, while for the Qwen2.5-VL models we use 8,192 tokens.

For evaluation, different protocols are used depending on the benchmark. In MM-Vet, we employ `gpt-4-0613` as the evaluator to score model responses. In CV-Bench and MMMU experiments, we adopt a rule-based string matching parser to determine correctness.

For SigLip2, we use the `siglip2-so400m-patch14-384`. The token cost reported in our tables is measured using the `tiktoken` library with the `cl100k_base` encoding.

It is worth noting that in the *img2svg* experiments, StarVector cannot take textual prompts as input. It directly performs image-to-SVG generation.

---

**Algorithm 1** Test-time Revision

---

1: **Input:** Coder $\psi$, an image $\mathcal{V}$, initial rendering $\widetilde{\mathcal{V}}^{(0)}$, initial SVG code $\mathcal{C}^{(0)}$, iteration number $T$
2: **Output:** Refined rendering $\widetilde{\mathcal{V}}^{(T)}$
3: **for** $t = 0 \to (T-1)$ **do**
4:      Comment the difference: $\Delta^{(t)} \leftarrow \psi\left(\mathcal{V}, \widetilde{\mathcal{V}}^{(t)}\right)$
5:      Generate revised SVG code: $\mathcal{C}^{(t+1)} \leftarrow \psi(\mathcal{V}, \widetilde{\mathcal{V}}^{(t)}, \Delta^{(t)}, \mathcal{C}^{(t)})$
6:      Update reconstruction: $\widetilde{\mathcal{V}}^{(t+1)} \leftarrow \text{Render}(\mathcal{C}^{(t+1)})$
7: **end for**
8: **return** $\widetilde{\mathcal{V}}^{(T)}$

---

## B   EXPERIMENTS ABLATIONS

**Effects by SigLip v.s. DINO**

| Metric | Claude-4-Opus | Claude-4-Sonnet | GPT-5 | GPT-4o | GPT-o3 | GPT-4.1 | GPT-4o-mini | Gemini-2.5-Pro | Gemini-2.5-Flash | Seed-1.6-thinking |
|---|---|---|---|---|---|---|---|---|---|---|
| SigLip2 | 67.2 | 66.9 | **70.1** | 60.1 | 64.9 | 66.9 | 58.6 | 66.4 | 64.3 | 62.6 |
| DINO-V2 | 26.1 | 24.2 | **30.4** | 16.5 | 22.4 | 26.0 | 14.3 | 27.2 | 22.5 | 20.2 |

Table 5: **Effect by different feature extractor** As shown, the DINO reach lower score compare with SigLip2, as it more focus on low-level visual representation. While SigLip2 focus on more on semantic space.

**Effect by revision on MM-Vet**

**Effects by different policy during evaluation on MM-Vet**

| Models | Round | Rec | Ocr | Know | Gen | Spat | Math | Avg. |
|---|---|---|---|---|---|---|---|---|
| Claude-4-Opus | 0 | 30.4 | 52.3 | 13.9 | 18.5 | 49.5 | 50.4 | 37.5 |
| | 1 | 29.0 | 54.3 | 18.9 | 21.7 | 56.0 | 53.1 | 38.8 |
| | 2 | 29.6 | 54.0 | 16.9 | 14.5 | 53.1 | 55.4 | 39.5 |
| GLM-4.5V | 0 | 22.4 | 54.4 | 7.1 | 15.6 | 46.0 | 56.9 | 33.1 |
| | 1 | 26.5 | 58.3 | 14.5 | 20.0 | 54.4 | 50.0 | 37.4 |
| | 2 | 24.5 | 65.7 | 10.4 | 15.6 | 53.3 | 55.8 | 38.3 |
| GPT-4o | 0 | 23.1 | 58.4 | 12.7 | 17.0 | 51.3 | 60.4 | 35.0 |
| | 1 | 23.7 | 53.5 | 12.0 | 15.7 | 46.5 | 53.5 | 34.1 |
| | 2 | 25.0 | 60.0 | 14.2 | 18.9 | 50.7 | 56.9 | 36.3 |

Table 6: **Effect by revision (round) on MM-Vet**.

| Setting | Evaluator | Rec | Ocr | Know | Gen | Spat | Math | Avg. |
|---|---|---|---|---|---|---|---|---|
| Ori | GPT-4o-mini | 60.5 | 78.9 | $\underline{58.5}$ | 59.5 | 70.9 | $\underline{84.2}$ | $\underline{67.1}$ |
| | Human | 41.7 | 69.4 | 18.6 | 23.1 | 70.1 | 73.8 | 51.6 |
| | Claude-4-Opus | **68.1** | $\underline{79.3}$ | **59.0** | $\underline{57.9}$ | **82.1** | 72.7 | 72.4 |
| | GLM-4.5V | $\underline{67.4}$ | **87.1** | 56.5 | **60.0** | $\underline{80.0}$ | **96.2** | **75.5** |
| VCoder | GPT-4o-mini | 46.0 | 64.4 | 40.8 | 43.0 | 61.6 | 72.7 | 54.1 |
| | Human | 26.3 | 48.3 | 13.7 | 14.4 | 47.9 | 55.0 | 35.9 |
| | Claude-4-Opus | 43.7 | 76.3 | 37.0 | 41.2 | 68.4 | 76.5 | 55.8 |
| | GLM-4.5V | 54.3 | 73.6 | 48.2 | 49.0 | 73.6 | $\underline{84.2}$ | 63.3 |

Table 7: **Evaluation results of different evaluators on Ori vs VCoder**

## C  PROMPT TEMPLATE

**Img2SVG**

```python
def _build_user_prompt():
    return """Convert this image to SVG code. Follow these rules:

CRITICAL REQUIREMENTS:
- Output only pure SVG code, no markdown blocks or explanations
- Start with <svg viewBox="..." xmlns="http://www.w3.org/2000/svg">
    and end with </svg>
- Use only native SVG elements (no external images or links)
- Include viewBox to ensure all elements are visible and auto-scale
    properly
- Calculate appropriate viewBox dimensions to contain all content
  with some padding

Generate the SVG now:"""
```

**Img2Text2SVG**

```python
def _build_user_prompt_stage1():
    return """Please provide a detailed and accurate description of
        this image. Focus on:

1. Main objects, shapes, and elements
2. Colors, textures, and visual properties
3. Spatial relationships and positioning
4. Style and artistic characteristics
```

```
5. Any text, symbols, or specific details

Be precise and comprehensive - this description will be used to
    recreate the image as an SVG. Include geometric details,
    proportions, and layout information that would be necessary for
    accurate reproduction."""

def _build_user_prompt_stage2(description):
    return f"""Based on the following description, generate clean
        and accurate SVG code:

{description}

Requirements:
1. Output ONLY complete SVG code, no explanations or other text
2. Use appropriate dimensions (e.g., viewBox="0 0 400 400" or
    similar)
3. Include all elements described with accurate colors, shapes, and
    positioning
4. Use clean, well-structured SVG syntax
5. Ensure the SVG is self-contained and complete
6. Start with <svg and end with </svg>
7. Use precise geometric shapes and paths where appropriate
8. Match colors and proportions as closely as possible to the
    description

Generate the SVG now:"""
```

**Img2SVG-***Thinking*

```
def _build_user_prompt():
    return """Let's analyze this image and create an SVG
        representation through a structured thinking process.

Step-by-step analysis:
1. Visual Decomposition
- What are the main visual elements?
- What geometric shapes can be identified?
- What are the key colors and their relationships?

2. Structural Analysis
- How are elements arranged and layered?
- What are the proportions and spatial relationships?
- Are there any repeating patterns or symmetry?

3. SVG Implementation Strategy
- Which SVG elements best represent each component?
- What's the optimal drawing order?
- How to handle complex shapes and gradients?

4. Technical Considerations
- What viewport dimensions are appropriate?
- How to ensure scalability and responsiveness?
- What optimizations can be applied?

After your analysis, provide:
1. Your complete reasoning process
2. The final SVG code implementation

Requirements for SVG output:
```

```
      – Must be complete and self-contained
      – Include all necessary attributes and elements
      – Start with <svg tag and end with </svg>
      – Use appropriate viewBox and dimensions

      Please proceed with the analysis and generation:"""
```

**All In One**

```python
def _build_system_prompt():
    return """You are a helpful assistant that converts images into
        clean, complete SVG vector graphics.

Your primary task is img2svg conversion for Visual Question
    Answering. You have access to two types of metadata to assist
    with precision:

METADATA AVAILABLE:
– OCR metadata: Text regions with precise 4-point quadrilaterals
    for accurate text placement
– Object detection metadata: Object boundaries with labels,
    confidence scores, and svg_path outlines

SPECIAL CASE HANDLING (Hint Strategy):
Sometimes, an image may depict a person, character, or artwork
    where fine details like facial features or texture could be lost
     during vectorization. Examples include:
– A recognizable public figure such as a scientist or political
    leader
– A well-known fictional character from popular culture
– A famous painting or portrait by a specific artist

If the subject in the image is of this nature and important
    identity cues might be lost:
– Preserve recognizability by including visual hints such as
    characteristic clothing, accessories, environment, or symbolic
    elements
– When confident, you may add a <text> element near the subject
    that provides:
        Their commonly known name
        The name of the associated work or series
        The title or creator of an artwork

If the subject does not fit these examples or is not clearly
    recognizable:
– Generate a clean SVG with no extra text labels
– Focus on accurate shapes, proportions, and composition

METADATA INTEGRATION:
1) Text rendering: Use OCR quadrilaterals as authoritative
    coordinates for text placement. Render literal text strings with
     appropriate transforms for rotation/skew.
2) Object boundaries: Use detection svg_paths as authoritative
    contours. Infer fill/stroke colors and add internal details
    within these boundaries.
3) Background reconstruction: Fill in unlabeled regions using
    native SVG primitives.

PROCESSING PRIORITY:
1. Use provided metadata for precise positioning (OCR quads,
    detection paths)
```

```
2. Apply hint strategy for recognizable subjects
3. Reconstruct missing background/unlabeled areas
4. Ensure proper layering and visual completeness

OUTPUT REQUIREMENTS:
- Output only pure SVG code, no markdown blocks or explanations
- Start with <svg viewBox="0 0 {W} {H}" xmlns="http://www.w3.org
    /2000/svg"> and end with </svg>
- Use only native SVG elements (no external images or links)
- Include viewBox to ensure all elements are visible and auto-scale
     properly
- Do not include explanations or commentary

This SVG will be used in a Visual Question Answering task, so
    ensure the output retains as much semantic identity as possible
    when visual details are reduced."""

def _build_user_prompt():
    return f"""METADATA:\n{metadata_json}\n\nGenerate the complete
        SVG with precise metadata integration and appropriate hint
        strategy for recognizable subjects."""
```

**Revision**

```
def _build_user_prompt_stage1():
    return """Compare the original image (first) with the SVG-
        rendered image (second) and identify SPECIFIC differences
        for SVG code revision.

Focus on identifying:

1. LOCATION-SPECIFIC DIFFERENCES:
   - Which areas/regions differ (top-left, center, bottom-right,
       etc.)
   - Missing or extra elements in specific positions

2. VISUAL ATTRIBUTE DIFFERENCES:
   - Color mismatches (specify which elements and what colors)
   - Shape distortions (which shapes are wrong and how)
   - Size/proportion issues (which elements are too big/small)
   - Position/alignment problems

3. SPECIFIC SVG REVISION SUGGESTIONS:
   - Which SVG elements need modification (circles, paths, rects,
       etc.)
   - What attributes to change (fill, stroke, cx, cy, width, height
       , d, etc.)
   - Specific color values or coordinate adjustments needed

Format your response as actionable SVG revision instructions."""

def _build_user_prompt_stage2():
    return """You are an SVG code specialist. Based on the visual
        analysis and comparison between the original image and
        current SVG rendering, make SPECIFIC code modifications to
        fix identified issues.

VISUAL ANALYSIS FINDINGS:
{optimization_goals}

CURRENT SVG CODE:
```

```
{current_svg_code}

INSTRUCTIONS:
1. Analyze the current SVG code structure and elements
2. Based on the visual analysis findings, identify which specific
    SVG elements need modification
3. Make precise changes to fix the identified issues:
   – Adjust colors (fill, stroke attributes)
   – Fix shapes and paths (modify d attributes, coordinates)
   – Correct sizes and positions (width, height, cx, cy, x, y)
   – Add missing elements or remove incorrect ones
4. Output ONLY the complete revised SVG code
5. Ensure all modifications directly address the issues mentioned
    in the analysis
6. Start with <svg and end with </svg>

Revised SVG code:"""
```

## D  MORE VISUALIZATION

## E  THE USE OF LARGE LANGUAGE MODELS

We used large language models (LLMs) in two limited ways. First, during manuscript preparation, LLMs were employed solely for surface-level editing (*e.g.,* grammar correction and minor rephrasing) to improve readability; they were not used to generate research ideas, methods, experiments, or conclusions. Second, in our benchmark experiments, LLMs were included as baseline models for comparison, with results reported transparently in the main paper. All core research contributions, dataset design, and analyses are the sole work of the authors.

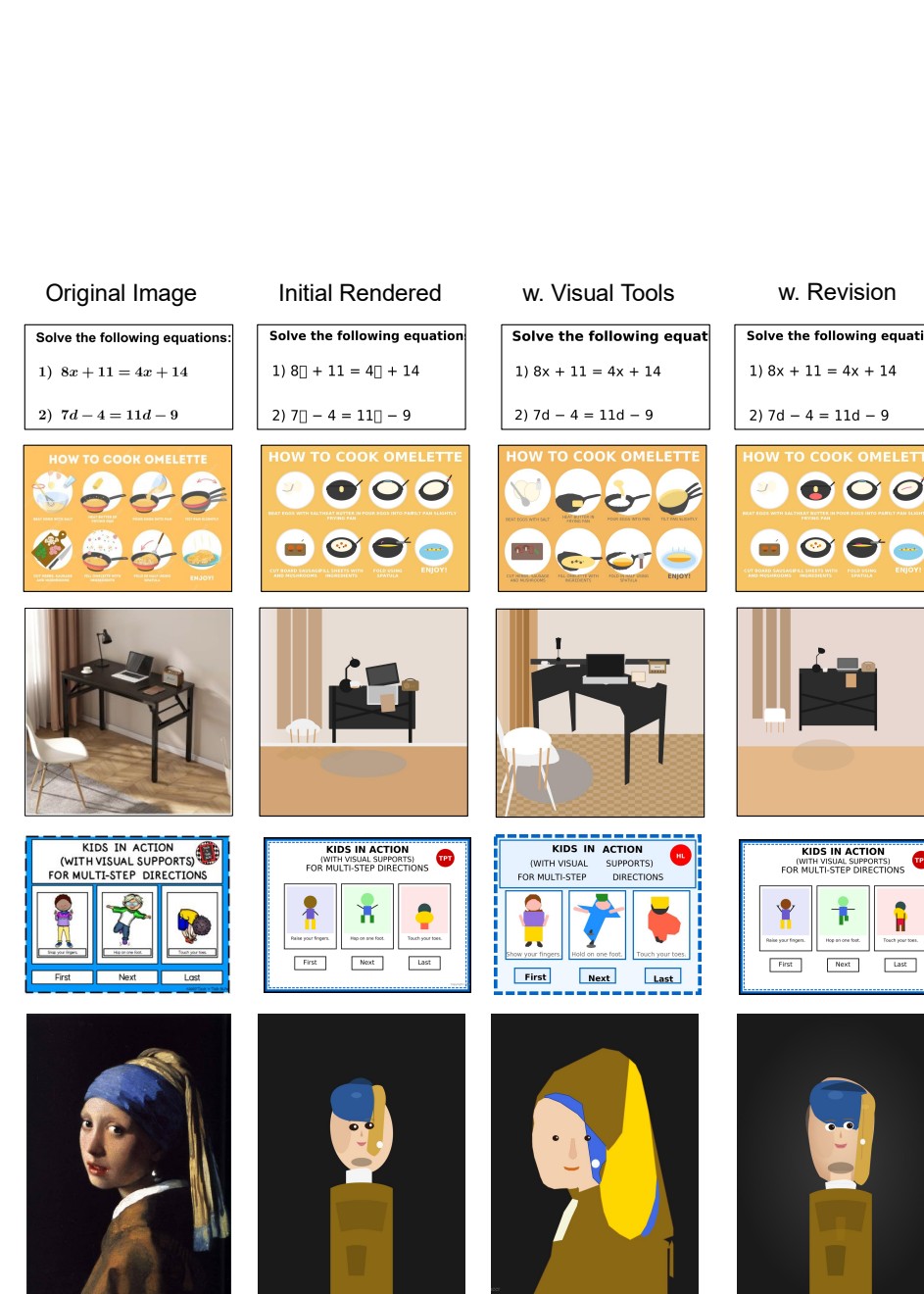

Figure 7: **More visualization results**

