# OpenReview forum: "VCode: a Multimodal Coding Benchmark with SVG as Symbolic Visual Representation"
_ICLR.cc/2026/Conference — Submitted to ICLR 2026_

### Official Review · Reviewer_v2u8 · 2025-10-26

**Soundness:** 3
**Presentation:** 3
**Contribution:** 3
**Rating:** 6
**Confidence:** 3

**Summary:**

This paper introduces VCode, a benchmark for visual-centric multimodal coding that redefines multimodal understanding as the task of generating SVG code from images. The work is motivated from the observation that most multimodal and coding benchmarks focus on linguistic or pixel-based representations, whereas SVG provides a symbolic, interpretable, and executable visual abstraction.
VCode covers three domains:  1) General commonsense (MM-Vet), 2) Professional knowledge (MMMU), and 3) Visual perception (CV-Bench).
To evaluate the symbolic fidelity of SVG representations, the authors propose CodeVQA, where a policy model must answer questions about the rendered SVG image.
They further introduce VCoder, an augmented agentic framework that enhances existing vision–language models (VLMs) through:
1) Thinking with revision: iterative refinement based on visual discrepancies between generated and target images
2) Acting with visual tools: using detectors, segmenters and OCR to provide structured cues like object boundaries with text.

Empirical results show that leading VLMs (e.g., GPT-5, Claude-4-Opus, Gemini-2.5) struggle on visual coding tasks, while VCoder achieves an +8.7-point overall improvement over Claude-4-Opus. Human studies show that people reason more robustly over symbolic SVGs than raw images, suggesting that symbolic visual coding could be key to more human-like multimodal intelligence

**Strengths:**

1) The paper introduces a novel paradigm: treating image understanding as code generation (SVG rendering).
2) The VCoder framework combining iterative refinement and external visual tools aligns with recent trends in agentic model enhancement.
3) Experiments are comprehensive, covering both closed- and open-source VLMs with detailed ablations (revision loops, tool usage, modality inputs).

**Weaknesses:**

1) The dataset contains only 464 image–question pairs, which is small compared to major multimodal benchmarks. Although the repurposing from MM-Vet/MMMU/CV-Bench ensures diversity, it may limit generalization and statistical reliability of reported differences.
2) CodeVQA uses an external policy model (GPT-4o-mini) as evaluator. This introduces evaluation bias and circularity, especially since some tested models are from the same family.
3) While the paper argues that SVG captures symbolic abstraction, it lacks quantitative or theoretical grounding for what constitutes “symbolic fidelity.” E.g., there could be metrics for structural alignment (e.g., object counts, relative positions) alongside SigLip and VQA accuracy.

**Questions:**

1) How sensitive are the CodeVQA scores to the choice of the policy model? Would results differ significantly if another evaluator (e.g., Claude-Sonnet or Gemini-Pro) were used?
2) Why was SVG chosen over other symbolic representations like scene graphs or DSLs for vector graphics? Could the same paradigm extend to 3D symbolic representations (e.g., CAD or mesh code)?
3) In Table 4, the Img2Text2SVG pipeline outperforms direct Img2SVG. Does this suggest that current models inherently reason better through language than through direct visual coding?

---

> ### Author Response · Authors · 2025-11-20
> **Author's Response to Reviewer v2u8**
>
> # W1: Limited dataset size and uncertainty about generalization.
> Our scale is reasonable because **each sample requires long-context code generation**, unlike short-answer benchmarks:
>
> | Type | Output | Avg. Output Length (tokens) | Avg. Time Cost (sec) | Purpose |
> |----------------|----------------|---------------------|---------------------------|----------|
> | MM-Vet, MMMU | Multiple-choice label / short phrase | 1–100 | 0.02–1 | Recognition or Reasoning |
> | VCode | Executable code | 500–3000+ | 250 | Symbolic visual coding |**
>
> Since each VCode sample is far more costly to produce and evaluate, 464 symbolic instances are appropriate for a first benchmark in this regime.
>
> Regarding generalization reliability, our results remain consistent across multiple evaluators and humans. As shown in **Fig. 4 and Tab. 7 (Appendix B)**, *changing the policy model or using human evaluation preserves the relative ranking of baselines, indicating that the VCode is stable despite its moderate size.*
>
> # W2: Risk of biased or circular evaluation due to GPT-4o-mini.
> We acknowledge possible evaluator bias. However, GPT-4o-mini is robust at perceiving whether an image reflects certain points, and the goal of CodeVQA is **not to** measure the policy itself, but to **test whether the rendered SVG contains sufficient visual semantics to answer correctly**. Similar to human evaluation, some degree of subjectivity is unavoidable, but the key question (**whether the representation is interpretable) remains meaningful.**
>
> To further reduce evaluator bias, we test **alternative policy models**, and both Fig. 4 and Table 7 (Appendix) confirm that *changing the evaluator does not alter the performance ranking of the tested models*. This suggests that CodeVQA is robust, rather than artifacts of a particular evaluator family.
>
> # W3: Absence of a principled metric to assess the SVG fidelity.
>
> To provide additional metrics for SVG symbolic fidelity, we add several image-level structural metrics computed between the rendered SVG and the original image.
> - **SSIM (Structural Similarity)**: evaluates structural layout and luminance/contrast consistency.
> - **LPIPS (Perceptual Metric)**: measures perceptual similarity using deep features, capturing higher-level structural alignment.
> - **Aesthetic Score**: approximates human aesthetic preference, reflecting overall visual coherence.
>
> These metrics quantify structural and perceptual fidelity beyond SigLIP and CodeVQA. As shown below, trends are **consistent** across metrics.
>
> | Model             | SigLIP↑ | CodeVQA↑ | SSIM↑| LPIPS↓| Aesthetic↑|
> |-------------------|--------|--------|-----------|-----------|-----------|
> | VCoder            | 71.0 | 54.0| 61.4 | 68.0 | 4.73      |
> | GPT-5             | 72.3 | 46.8 | 58.1 | 71.9 | 4.96      |
> | Claude-4-Opus     | 65.9 | 41.7| 54.8 | 73.7 | 4.78      |
> | Qwen2.5-VL-72B    | 57.9 | 36.0 | 42.7 | 79.3 | 4.36      |
> | GLM-4.5V          | 63.8 | 40.1 |49.9 | 77.5 | 4.57      |
>
>
> # Q1: Sensitivity of CodeVQA to the choice of evaluator model.
> We have already examined this. As shown in Fig. 4 and Table 7 (Supp), replacing GPT-4o-mini with **alternative policy models** (e.g., Claude-4-Opus, GLM-4.5V, Human-eval) yields highly consistent performance ordering across evaluated models. This indicates that **CodeVQA scores are not overly sensitive to the particular choice of evaluator**.
>
> # Q2: Why is SVG preferred, and is the paradigm applicable to 3D symbolic representations?
> We chose SVG because it is the **most widely used, concise, and easily parseable symbolic format** for 2D visual content. Compared with scene graphs or custom DSLs, SVG offers several practical advantages:
> 1. It has a **simple and standardized syntax**, without the heavy engineering rules required by DSLs or professional complex syntax (such as HTML, CSS).
> 2. It is **natively supported across browsers and rendering engines**, making it easy to validate and visualize;
> 3. Recent **frontier VLMs (e.g., Gemini 3.0) widely use SVG in their demos as testbed** for structured visual outputs, showing that SVG is consistent with current model capabilities and usage patterns.
>
> Regarding 3D symbolic representations such as CAD or mesh code, the same paradigm is conceptually applicable. We view this as **a promising direction for future extensions.**
>
> # Q3: Does the Img2Text2SVG indicate a preference for text-centric reasoning?
> Exactly! This result is expected and aligns with our motivation. Current VLMs generally have stronger reasoning ability in the **language-query** rather than in direct **visual-query**.
> The gap observed between Img2SVG and Img2Text2SVG highlights that **direct visual coding remains an underexplored capability** and that existing models rely heavily on linguistic media.
> This *reinforces the value of VCode as a complementary testbed for evaluating **visual-centric symbolic generation**.*

---

> > ### Author Response · Authors · 2025-11-24
> > **Gentle Reminder to Review Our Rebuttal**
> >
> > Dear Reviewer v2u8,
> >
> > Thank you once again for your feedback!
> >
> > We would greatly appreciate it if you could review our response to ensure it adequately addresses your concerns. We remain fully dedicated to clarifying any remaining points and would welcome any further discussion to ensure all your questions are thoroughly answered.
> >
> > Thank you for your time and consideration.
> >
> > Best,
> >
> > Authors of 5638

---

### Official Review · Reviewer_j8Bz · 2025-11-01

**Soundness:** 2
**Presentation:** 2
**Contribution:** 2
**Rating:** 0
**Confidence:** 5

**Summary:**

The paper introduces VCode, a benchmark framing multimodal understanding as generating SVG code from images and reasoning over the rendered output. It proposes CodeVQA to test whether SVG-based representations preserve semantic visual information and VCoder, which combines iterative code revision and vision-tool assistance. Experiments show gains over existing VLM coders but also reveal persistent weaknesses in fine-grained visual reasoning.

**Strengths:**

1. The idea of using SVG as an intermediate symbolic space for vision-language reasoning is conceptually novel and touches on an underexplored direction in multimodal representation.

2. The work incorporates test-time revision and tool-assisted perception, which reflects awareness of limitations in current models and attempts to address them through modular augmentation rather than purely scaling.

**Weaknesses:**

1. The evaluation protocol is fragile: SigLIP similarity offers weak guarantees on fine-grained structure, and CodeVQA depends on the answering model’s biases and failure modes, making correctness a function of the evaluator rather than the representation. This undermines reliability and fairness, which is critical for a benchmark.

2. The dataset is almost entirely repurposed from prior benchmarks without substantial new curation or justification for domain coverage, scale, or annotation quality. As a result, it is unclear whether the benchmark truly captures the core challenges of the proposed problem.

3. The approach lacks grounding in practical vision tasks or downstream applications, and the SVG abstraction remains unconvincing as a scalable representation (especially for natural images with complex textures, occlusions, or fine geometry). Additional empirical evidence or ablations are needed to justify that benefits outweigh the loss of fidelity and that this direction can generalize beyond small synthetic-like cases.

**Questions:**

Is the evaluation protocol reliable?

---

> ### Author Response · Authors · 2025-11-20
> **Author's Response to Reviewer j8Bz**
>
> > We note that your main concern lies in the **reliability of the evaluation protocol**, and we would like to address each of your points in turn.
>
> # W1: Fragility of evaluation caused by SigLIP sensitivity and CodeVQA model bias
> Thank you for raising this important concern. As VCode is a new visual-centric coding setting, there is no existing perfect metric. To increase reliability, our protocol intentionally combines **multiple complementary dimensions**:
> 1. code-only metric (SVG success rate, code length),
> 2. vision-only metric (SigLIP and DinoV2 score), and
> 3. symbolic understanding via coding (CodeVQA).
>
> This multi-view evaluation reduces dependence on any single metric.
>
> Regarding **SigLIP**, we compared it with DinoV2 in Tab5 (Appendix), which focuses more on fine-grained structure. However, we found that DINOv2 is hard to differentiate model behavior; its scores are mostly clustered, while SigLIP embedding focus on semantic is better. This show that **fine-grained signal is not the primary factor in SVG images**. **Our goal is to capture semantic consistency, not low-level appearance matching.** There is why we propose CodeVQA as a new metric.
>
> For **CodeVQA**, it based on a evaluator to connect coding and symbolic understanding.
> In Fig. 4 and Tab. 7 (Appendix), we validate that **changing the evaluator does not alter the performance ranking of the tested models** and observed strong alignment, indicating that it acts as a reliable semantic verifier. Similar evaluator-based approaches **(VLM-as-a-judge)** are widely adopted in multimodal benchmarks despite inherent subjectivity.
>
> Overall, by combining three signals jointly, we ensure that VCode provides a fresh and reliable evaluation of visual-centric SVG representations.
>
> # W2: Whether the repurposed dataset’s coverage matches the proposed task
> Although our dataset is repurposed from existing benchmarks, **the selection is non-trivial**. We intentionally choose three complementary domains—**general reasoning** (MM-Vet), **college-level knowledge** (MMMU), and **visual-centric perception** (CV-Bench). Because *they represent the broadest and most widely adopted evaluation axes for foundation VLMs*, collectively provide:
>
> 1. **Broad coverage:** matching the scope and dimension commonly reported by top-performing multimodal models.
> 2. **Diverse visual challenges:** ensuring the generated SVGs must encode both symbolic and perceptual information.
> 3. **High-quality annotations:** the question-answering pairs are already validated by the human and community.
>
> We want to emphasize that VCode, as **the first benchmark** to study and  bridge visual-centric SVG representation and multimodal reasoning, **uses these well-established datasets as a consistent and reliable starting point**. Our contribution lies in introducing a new representation paradigm and a novel evaluation protocol, and these datasets are essential for establishing fair comparisons across VLMs.
>
> # W3: Doubts about the practicality and generalization of SVG as a scalable visual representation
> Thank you for the thoughtful comment!
> We clarify that VCode is intentionally designed as a **symbolic-centric representation benchmark, rather than a pixel-fidelity reconstruction task**. The goal explicitly stated in the paper (`line54-74`) is to evaluate whether a VLM can represent symbolic (objects, layouts, relations), not fine textures or photorealistic details.
>
> Notably, the three source benchmarks we adopt (MM-Vet, MMMU, CV-Bench) already **contain challenging visual phenomena** such as *spatial relations, counting, occlusion, and scene reasoning*. If SVG abstraction failed to retain essential semantics, the downstream CodeVQA results would collapse. Instead, our experiments show that models can reliably recover these structured cues from the SVG renderings, *indicating that fidelity loss does not hinder the core reasoning signals.*
>
> Moreover, *structured and executable outputs are increasingly central* to real multimodal applications. Such as recent frontier VLM (e.g., **Gemini3**) highlight the agentic and coding ability, and coding for symbolic representations are more practical than raw pixel fidelity. SVG provides one of the few formats that is both **visually interpretable and executable**, aligning well with these emerging use cases.
>
> We conducted an experiment to evaluate **Image classification on 100-class Mini ImageNet**. We sampled one image per class, converted it to SVG using VCoder, and evaluated recognition using `GPT-4o-mini` as the classifier. The classifier achieves 72% accuracy (original image with 92%), showing that SVG representations as a valid representation to preserve semantic structure for images.
>
> As the **first benchmark for visual-centric symbolic representation**, our goal is to establish a problem definition and provide a reproducible evaluation pipeline. We believe this remains valuable even if full coverage of real-world visual complexity is left to future work.

---

> > ### Author Response · Authors · 2025-11-24
> > **Gentle Reminder to Review Our Rebuttal**
> >
> > Dear Reviewer j8Bz,
> >
> > Thank you once again for your feedback!
> >
> > We would greatly appreciate it if you could review our response to ensure it adequately addresses your concerns. We remain fully dedicated to clarifying any remaining points and would welcome any further discussion to ensure all your questions are thoroughly answered.
> >
> > Thank you for your time and consideration.
> >
> > Best,
> >
> > Authors of 5638

---

> > > ### Comment · Reviewer_j8Bz · 2025-11-28
> > >
> > > I thank the authors for their detailed rebuttal. I have read the response and the additional experimental results.
> > >
> > > While I appreciate the clarification on the practicality of SVG representations (W3), which I now consider addressed, I remain unconvinced by the responses regarding the novelty of the setting (W2) and the reliability of the metrics (W1). Consequently, the fundamental flaws in the motivation and evaluation design prevent me from recommending acceptance.
> > >
> > > Here is a detailed breakdown of why the rebuttal arguments for W1 and W2 are insufficient:
> > >
> > > 1. The Claim of Novelty is Contradicted by Cited Works (Re: W2) The rebuttal states that "VCode is a new visual-centric coding setting" and claims to be "the first benchmark to study and bridge visual-centric SVG representation and multimodal reasoning."
> > >
> > > This claim is factually overstated and contradicts the authors' own Related Works section. The task of "Visual-centric coding" (specifically Image-to-SVG generation) has been established by prior works that are cited in this very paper:
> > >
> > > - SVGenius: Explicitly benchmarks LLMs in "SVG understanding, editing and generation."
> > >
> > > - StarVector: Titled "Generating scalable vector graphics code from images and text," which directly addresses the visual-centric coding setting.
> > >
> > > - OmniSVG: A unified model specifically for SVG generation.
> > >
> > > Therefore, characterizing VCode as the "first" to bridge this gap is inaccurate. While VCode introduces a new evaluation protocol (CodeVQA), the task setting itself is not novel. The paper should position itself as an evaluation refinement, not a task inception.
> > >
> > > 2. The Unreliability of the Evaluation Metrics (Re: W1) The rebuttal argues that comparing SigLIP vs. DINO demonstrates rigor. However, this misses the core concern regarding the validity of these metrics for this specific modality.
> > >
> > > - Embedding Metric Reliability (SigLIP): My concern is not about which model is used (SigLIP vs. DINO), but the fundamental validity of using cross-modal embedding similarity between Natural Images and Abstract Vector Renders. There is a massive domain gap here. A low cosine similarity score may simply reflect that one image has texture and the other does not, rather than a lack of semantic fidelity. Conversely, a high score might reflect color histogram matching rather than structural correctness. The rebuttal does not address how this domain shift is mitigated.
> > >
> > > - CodeVQA and "Shared Blindness": The rebuttal claims CodeVQA is reliable because rankings are consistent. However, Figure 4  and Table 7  reveal a critical "Shared Blindness" (or collusion) issue that invalidates this metric as a proxy for human-aligned visual quality.
> > >
> > > - Human < AI Anomaly: In the VCoder setting (evaluating rendered SVGs), Humans achieve a score of 40.6, while models like GLM-4.5V achieve 54.1. When a metric claims that a model understands the visual fidelity of an abstraction significantly better than a human, it strongly suggests the metric is measuring "model-to-model correlation" rather than true visual interpretability. The evaluator (GPT-4o-mini) likely hallucinates correctness based on shared statistical patterns with the generator that humans cannot perceive.
> > >
> > > - Evaluator Ceiling Effect: The benchmark includes challenging perception tasks. Using GPT-4o-mini, a smaller model with known limitations in complex visual reasoning, as the judge creates a validity ceiling. If the evaluator itself struggles with the spatial relations in the original images, it cannot reliably judge the preservation of those relations in the SVG abstractions.

---

> > > > ### Author Response · Authors · 2025-12-01
> > > >
> > > > We thank the reviewer for the continued engagement and glad that the concern regarding practicality of SVG is addressed.
> > > >
> > > > We understand your remaining concerns regarding novelty and metric reliability. We would like to offer individual clarifications about experimental designs that strongly suggest these are not ‘fundamental flaws,’ but rather deliberate design choices aligned with the specific goals of visual-centric symbolic representation.
> > > >
> > > > 1. **Novelty: from simple icons to complex natural scenes (Re:W2)**
> > > > While we cite *SVGenius, StarVector,* and *OmniSVG*, we want to highlight why VCode as the first visual-centric in the context of **domain and evaluation focusness**.
> > > > - **Prior Work (Graphics focus + reconstruction-level similarity):** Existing works focus on generating simple, synthetic assets like icons, logos, or charts. The visual complexity is low, and the goal is aesthetic generation. Prior generative benchmarks do not address and just focus on measure the reconstruction level difference.
> > > > - **VCode (Natural scene + symbolic reasoning):** We are the first to bridge SVG representation with **natural images**, which is more general, in-the-wild. Converting a complex photo (with occlusion, texture, depth) into SVG for reasoning is a fundamentally harder and different task than icon generation. We believe that pixel metrics work for icons but fail for **abstracted natural scenes**. This shift necessitates our new protocol.  There is why CodeVQA is specifically designed to bridge this “Natural Image -> Symbolic Code” translation,
> > > >
> > > > 2. **SigLIP: Measuring semantic consistency, Not pixel fidelity  (Re:W1)**
> > > > You raised valid concerns about the domain gap affecting SigLIP.
> > > >
> > > > - **Motivations behind SigLIP**: Our goal is **Symbolic representation**, not "compression for reconstruction." We do not aim for the SVG to look pixel-perfectly identical to the photo. Instead, we aim for **Semantic Consistency**, preserving the layout, objects, and relations in a lightweight, editable format. There is why we think about SigLIP first.
> > > > - **SigLIP provide valid alignment** We report the difference among different visual encoders, showing that SigLip is better than DINOv2, as SigLip trained on massive web-scale images to align images with text semantics, which provides robuts embeddings even in SVG. While DINO focus on fidelity trained by self-supervised learning, which show worse alignment.
> > > > - **SigLIP still no perfect**: In our main body, we do not claim SigLIP is a perfect metric as it just focus on visual embedding consistency, it remain unclear so we develop CodeVQA.
> > > >
> > > > 3. **Explaining "Human < AI": The Knowledge gap vs. Hallucination (W1)**
> > > > The reviewer suspects "Shared blindness" (hallucination) because models outperform humans (54.1 vs. 40.6). We respectfully argue this gap stems from **knowledge**.
> > > > - **Experimental Setup:** Both Humans and Models receive the same pure visual input (rendered SVGs).
> > > > - **The Cause:** Many VCode tasks require expert domain knowledge (e.g., identifying a specific circuit component, historical architecture, or medical diagram).
> > > >     - **Humans:** A layperson human may visually see the SVG clearly but fail to answer the reasoning question due to a lack of domain knowledge (A graduate student might not understand archimedes' work)
> > > >     - **Models:** Large VLMs possess vast internal knowledge bases. Once they parse the visual structure (via the SVG), they can leverage this knowledge to answer correctly. (Seeing sunflowers can evoke images of Van Gogh.)
> > > > - **Conclusion:** The fact that models score higher indicates that the SVG representation successfully retains enough visual structure to trigger the model's correct reasoning capabilities. It reflects the **utility of the representation for AI agents**, which is the primary purpose of VCode.
> > > >
> > > > **4. Evaluator Ceiling & Reliability (Re: W1)**
> > > > - **Robustness:** We want to emphasis that GPT-4o-mini as a normal-level foundation vision model is strong enough to understand most common natural image. We hope that the rendered SVG images will be acceptable to most average students, not just a few exceptional ones.
> > > > - **Trend v.s. absolute value:** Our ablation studies (Fig. 4/Table. 7) show that **performance rankings remain consistent** across different evaluators. This proves that the metric reliably *distinguishes better representations from worse ones*, even if the absolute score is bound by the judge's capacity.
> > > > - **Upgrade the GPT-4o to frontier VLM:** To address the "ceiling" concern and ensure rigorous fairness, we are hapy to upgrade the primary evaluator from **GPT-4o-mini** to **GPT-5** in our revision experiments.

---

### Official Review · Reviewer_5CGi · 2025-11-01

**Soundness:** 2
**Presentation:** 3
**Contribution:** 3
**Rating:** 4
**Confidence:** 4

**Summary:**

This paper proposes a novel visual-centric benchmark that reframes multimodal understanding as code generation, and proposes an evaluation protocol that a policy model answers questions over rendered SVG. The paper finds that there is a persistent gap between language-centric and visual-centric coding, so it proposes an agentic framework that provides VLMs with two abilities: (1) thinking with revision; (2) acting with visual tools. The experimental results show that the proposed framework achieves a significant improvement in the visual-centric benchmark.

**Strengths:**

1. Extending language-centric coding to a new visual-centric coding task is an interesting and novel research direction.

2. This paper converts the multimodal understanding task into a visual-centric coding task and utilizes a Visual Model (VLM) to evaluate whether the generated code is an adequate and faithful visual representation.

3. The proposed VCoder framework is equipped with two capabilities: thinking with revision and acting with visual tools. Experimental results demonstrate the effectiveness of the proposed method.

**Weaknesses:**

1. The dataset in this paper was not processed; it simply used the original images and QA from MM-Vet, MMMU, and CV-Bench. Since the SVG code is entirely generated by the VLM being evaluated, the authors only proposed SVG code generation as a benchmark approach. This benchmark does not design a unified principle for SVG code generation to guide subsequent VLM generation. The lack of a unified principle for SVG code generation can easily lead to instability in the generated code, resulting in unstable code evaluation.

2. While CodeVQA evaluates the accuracy of code generation, I believe there are two issues: First, code generation itself is a capability that needs careful evaluation; it should not be confused with multimodal understanding. For example, minor issues with the SVG code might cause rendering failures, leading to poor results, but this does not necessarily mean the model's understanding is flawed. Second, CodeVQA is easily influenced by text input. When the policy model struggles to obtain effective information from the rendered image, it may output an answer based on publicly available knowledge from the text input.

3. CodeVQA is disadvantageous for small models (e.g., models with around 7 parameters or less) because these small models have difficulty generating compliant SVG codes, resulting in poor final evaluation results. However, in reality, these small models have already achieved very good results in multimodal understanding capabilities.

**Questions:**

1. Refer to the issues raised in the weaknesses section.
2. The authors lack experimental results on other baseline models.

---

> ### Author Response · Authors · 2025-11-20
> **Author's Response to Reviewer 5CGi**
>
> # W1: Lack of a unified SVG protocol causing instability.
> Thank you for the comments! We agree that stable SVG generation is challenging.
> To address this, we use three strategies:
> - **Standardize prompt template**
> All models follow the same SVG prompt (Appendix Sec. C), ensuring consistent generation across baselines.
> - **Post-processing SVG codes**
> We clean and validate predicted SVGs using regex rules to ensure they are fully renderable before evaluation.
> - **Automatic retry for failed cases**
> For any failed output, we allow up to 3 automatic regenerations so capable models can still produce valid SVGs.
>
> These steps ensure a reproducible pipeline and reduce instability in both code and evaluation.
>
> # W2: Whether CodeVQA mixes code validity with multimodal understanding or textual priors.
> Thank you for the insightful comments. We clarify that code ability and multimodal understanding are evaluated **separately**:
> - **Independent code-generation evaluation**
> As shown in Table 2, we report SVG Success Rate (SR) and Code Tokens (K), measuring only syntactic validity and compactness. This part does **not** involve multimodal reasoning. The SR variation across models shows that code generation itself is measurable and distinct.
> - **Multimodal score uses visual similarity**
> We use SigLIP to assess image-level similarity between the original and rendered images. However, these metrics alone **cannot** fully measure semantic preservation, motivating a more representative metric.
>
> - **CodeVQA connects visual coding with symbolic semantics**
> SVG generation is neither algorithmic (LeetCode) nor text-matching (having a unique answer).
> Since our outcome is visual, this requires a verifier to handle variances in visual outputs. CodeVQA therefore serves as a new prototype verifier by jointly considering the code-rendered image and symbolic  representation, enabling evaluation of whether the visual semantics are well preserved.
>
> - **CodeVQA is not driven by textual priors**
> To show that CodeVQA does not rely on textual priors or question leaking, we conduct a control experiment where **the rendered image is completely removed** and the evaluator receives **only the question and options**.
> Without any image input, the evaluator’s performance drops **far below random guessing**, indicating that textual information alone is insufficient and the model may be biased toward particular options.
>
> | Setup | 2D↑| 3D↑| Overall↑|
> |-----------|------|------|---------|
> | Random    | 36.8 | 50.0 | 43.4    |
> | Visual-Blind  | 4.3  | 23.3 | 13.8    |
> | VCoder  | 57.7 | 65.0 | 61.3|
>
> This confirms the **task requires visual input and avoids question leaking**.
>
> # W3: Are small models unfairly disadvantaged in CodeVQA?
> We clarify that VCode is **a new and under-explored setting**, and the evaluation itself is reasonable for all model sizes. To minimize format-related failures, we enforce a template prompt, post-processing, and automatic multi-try regeneration to minimize accidental syntax issues.
>
> Smaller models may still struggle to produce valid SVGs, but this reflects a **real capability gap** not captured by saturated multimodal benchmarks (eg MM-Vet). VCode targets emerging abilities like long-context, structured, and format-constrained visual coding are central for frontier VLM such as for agent tasks.
>
> Thus, difficulty for small models is **not** a flaw of the benchmark but highlights missing abilities that current evaluation fails to capture. This motivates VCode as **a timely and complementary testbed.**
>
> # Q2: Add results for other baseline models.
> Thank you for pointing this out. We added results for more baselines — `Gemini-3-Pro`, `Claude-4.5-Sonnet`, `Llama-4-Scout-17B-16E`, `Qwen3-VL-235B-A22B`, and specialist `OmniSVG` — broadening model coverage and comparison.
>
> |Model|Success Rate(%)|SigLIP|Code Token(K)|Rec|Ocr|Know|Gen|Spat|Math|CodeVQA-Avg|MMMU-Avg|2D|3D|CVBench-Avg|Overall|
> |-----|--------------|------|------------|---|---|----|----|----|----|-----------|--------|--|--|-----------|-------|
> |**Gemini-3-Pro**|100.0|74.4|1.8|45.6|72.7|32.6|36.9|71.9|80.0|55.4|42.5|59.1|58.3|58.7|52.1|
> |**Claude-4.5-Sonnet**|99.1|66.8|1.9|29.7|57.6|11.9|17.0|57.3|52.7|39.0|42.5|50.4|55.0|52.7|43.1|
> |**Llama-4-Scout-17B-16E**|100.0|55.5|0.7|18.2|44.9|12.4|15.5|32.8|46.2|26.4|42.5|35.0|53.3|44.2|35.3|
> |**Qwen3-VL-235B-A22B**|95.1|58.1|1.7|19.3|54.6|8.8|14.5|45.6|53.1|31.1|41.1|22.6|58.3|40.5|36.3|
> |**OmniSVG**|100.0|46.2|5.3|9.2|15.3|3.7|10.4|16.9|11.5|9.4|43.8|24.8|40.0|32.4|25.2|

---

> > ### Author Response · Authors · 2025-11-24
> > **Gentle Reminder to Review Our Rebuttal**
> >
> > Dear Reviewer 5CGi,
> >
> > Thank you once again for your feedback!
> >
> > We would greatly appreciate it if you could review our response to ensure it adequately addresses your concerns.
> > We remain fully dedicated to clarifying any remaining points and would welcome any further discussion to ensure all your questions are thoroughly answered.
> >
> > Thank you for your time and consideration.
> >
> > Best,
> >
> > Authors of 5638

---

### Author Response · Authors · 2025-11-20
**General Summary**

We sincerely thank all the reviewers for their time and constructive feedback. We are encouraged by the reviewers' recognition that:
* **VCode is being an interesting and novel direction** (5CGi, j8Bz, v2u8).
* **The proposed agentic framework is technically sound and motivated** (5CGi, j8Bz, v2u8).
* **Experiments effectively validate the approach** (5CGi, v2u8).

Please find our detailed responses to your specific questions below. For clarity, we use the following notations:
* **W** – Weakness
* **Q** – Question

---

> ### Author Response · Authors · 2025-11-27
> **To ACs: Reminder for reviewers' discussion**
>
> Dear ACs,
>
> We appreciate all the reviewers for their valuable suggestions. In response, we have carefully addressed each concern in detail.
>
> We kindly request your assistance in reminding the remaining reviewers to review our rebuttals and engage in the discussion.
>
> Best,
> Authors of 5638

---

### Meta-Review · Area_Chair_GHVo · 2026-01-09

**Summary:**

The reviewers' primary concerns are on the novelty of the proposed task and the reliability of the evaluation protocol. While the idea of using SVG as a symbolic representation for visual reasoning was found to be interesting (5CGi, j8Bz, v2u8), there were significant doubts about the benchmark. The most substantial unresolved concern proposed by Reviewer j8Bz, is the fundamental validity of the proposed CodeVQA evaluation metric. The paper's own results show that AI models outperform humans at answering questions based on the generated SVGs (Human < AI). This suggests that the VLM evaluator rewards outputs that correlate with the generator model's internal biases, rather than outputs that are genuinely faithful and human-interpretable symbolic representations. The authors attributing this gap to the AI's superior domain knowledge might point to a confounding variable: the evaluation mixes the quality of the visual representation with the evaluator's pre-existing knowledge, making it impossible to isolate and measure the former.

Additional concerns included the overstatement of novelty that requires refinement (which is not done in the revised version), as prior work has already explored image-to-SVG generation (j8Bz), and the potential for evaluation bias and ceiling effects by using a specific, non-frontier VLM as the judge (j8Bz, v2u8, 5CGi). While the authors added more experiments and metrics in their rebuttal, these did not address the core methodological flaw in their primary evaluation contribution.

**Reviewer Concerns:**

Concerns Addressed by the Rebuttal:
- *Practicality and Scalability of SVG* (Reviewer j8Bz): The authors successfully argued that the goal is symbolic abstraction, not pixel-perfect reconstruction. They provided a downstream classification experiment (Mini-ImageNet) to show that the generated SVGs retain sufficient semantic information for other tasks. Reviewer j8Bz acknowledged this point was addressed.
- *Lack of Diverse Fidelity Metrics* (Reviewer v2u8): The authors added structural and perceptual metrics (SSIM, LPIPS, Aesthetic Score) to provide a more multi-faceted evaluation of SVG fidelity, satisfying the reviewer's request.
- *Dataset Size Justification* (Reviewer v2u8): The authors provided a reasonable justification for the dataset's scale by highlighting the high computational cost and long-context nature of generating SVG code for each sample.
- *Influence of Textual Priors* (Reviewer 5CGi): The authors conducted a "visual-blind" control experiment showing that the evaluator's performance collapses without the rendered SVG, demonstrating that the visual information is necessary and the metric is not solely driven by textual cues in the question.

Outstanding Concerns:
- *Fundamental Reliability of CodeVQA* (Reviewer j8Bz; v2u8): This is the critical outstanding issue. The Human < AI performance anomaly strongly suggests that the CodeVQA metric is not a valid measure of human-interpretable symbolic fidelity. The authors' explanation of a "knowledge gap" introduces a severe confounder that undermines the benchmark's claim to be evaluating the representation itself. This core methodological flaw was not resolved.
- *Overstated Novelty* (Reviewer j8Bz): The paper's claim to be the "first" to bridge SVG representation and multimodal reasoning is an overstatement, as prior work (e.g., SVGenius, StarVector), cited by the authors themselves, has already established the image-to-SVG task. The paper's contribution is more an extension to complex scenes with a new evaluation protocol, and its positioning should be more precise.
- *Evaluator Ceiling and Bias* (Reviewer j8Bz; v2u8): Using a single, non-frontier model (GPT-4o-mini) as the primary judge introduces a performance ceiling and potential biases. While the authors showed consistent rankings with other evaluators, the validity of the absolute scores and the metric itself remains questionable, especially for challenging perception tasks that may exceed the evaluator's own capabilities.

**Reviewer Scores:**

The original scores are 0, 4, 6. Reviewer j8Bz, who provided the most critical feedback, remained firm on their initial score of 0 after a detailed discussion with the authors. While a score of 0 may be considered harsh given the paper's interesting direction, the unaddressed methodological flaws still stand. Based on the outstanding concerns, if all reviewers participate in the discussion, the overall score can still lean towords rejection.

---

### Decision · Program_Chairs · 2026-01-26

Reject